# Degradation Law and Service Life Prediction Model of Tunnel Lining Concrete Suffered Combined Effects of Sulfate Attack and Drying–Wetting Cycles

**DOI:** 10.3390/ma15134435

**Published:** 2022-06-23

**Authors:** Feng Lu, Haiyan Wang, Lichuan Wang, Kai Zhao, Junru Zhang

**Affiliations:** 1School of Emergency Management, Xihua University, Chengdu 610039, China; fenglu0901@foxmail.com; 2Key Laboratory of Transportation Tunnel Engineering, Ministry of Education, School of Civil Engineering, Southwest Jiaotong University, Chengdu 610031, China; wlc773747@126.com (L.W.); jrzh@home.swjtu.edu.cn (J.Z.); 3College of Transportation Science & Engineering, Nanjing Tech University, Nanjing 210009, China; zhaokai@njtech.edu.cn; 4China Railway 18 Bureau Group Co., Ltd., Tianjin 300222, China

**Keywords:** tunnel lining concrete, deterioration resistance coefficient, sulfate attack, drying–wetting cycles, service life prediction

## Abstract

The present study explored the degradation law and service life prediction of tunnel lining concrete with different mineral admixtures under coupled actions of sulfate attack (SA) and drying–wetting (DW) cycles. The deterioration resistance coefficient (DRC) of compressive strength and influence coefficients of sulfate concentration, mineral admixture content, water/binder (w/b) ratio, and curing regime on DRC were studied. After that, a new service life prediction model based on damage mechanics was developed and analyzed. Results show that, by increasing the DW cycles, the DRC first increases and then decreases. DRCs of Ordinary Portland cement (OPC), fly ash (FA), and ground granulated blast-furnace slag (GGBS) concrete linearly decrease with the increase of sulfate concentration, while the silica fume (SF) concrete displays a two-stage process; by increasing the admixture content, the DRCs of FA and GGBS concrete exhibit two distinct stages, while the SF concrete depicts a three-stage process; increasing the w/b ratio linearly decreases the DRC; the DRC of curing regime was sequenced as standard curing (SC) > fog curing (FC) > water curing (WC) > same condition curing (SCC). Based on the experimental results, the service life prediction model is applied and validated. The validation results show that the proposed model can accurately predict the lifetime of concrete with different mix proportions. Furthermore, it is found that the mineral admixture can effectively improve the lifetime of concrete, and the composite mineral admixture is more effective than a single mineral admixture in improving the lifetime of concrete.

## 1. Introduction

Sulfate ions widely exist in groundwater, seawater, sulfate soil, and salt lakes. The external sulfate attack (SA) has become one of the main factors that cause degradation and shortening of the service life of concrete structures [1,2]. SA is a complex physical and chemical process in concrete, which can lead to strength reduction, softening, and weight loss due to the cohesiveness loss of the cement hydration products [3,4,5,6,7]. Aggressive damages, mainly caused by SA, have been widely found in underground structures [8,9,10,11,12,13], offshore and marine facilities [14], and concrete sewer pipelines in wastewater infrastructure systems [15], etc. In recent years, the research group has conducted a large number of investigations. The results show that there are a lot of sulfate and other corrosive substances in the rock stratum in southwest China, and most of the investigated tunnels have sulfate attack diseases in varying degrees (Figure 1). SA will cause the degradation of tunnel lining structure, which seriously threatens the stability of the tunnel. More seriously, drying–wetting (DW) cycles could accelerate the degradation [16], especially the concrete structures in splash and tidal zones, or the underground structures situated in water table fluctuations [17,18,19,20]. Thus, it is of great significance to investigate the degradation law and service life prediction of tunnel lining concrete under coupled actions of SA and DW cycles.

Cement-based composites subjected to SA have been paid much attention since the 1960s. SA proceeds as sulfate ions enter the porous concrete and react with cement hydration products to form products such as ettringite, gypsum, or thaumasite. All of these mineral products can lead to expansion, cracking, and degradation of concrete [21,22,23]. Several experiments have been conducted to reveal the expansion and degradation mechanism of concrete under SA [24,25,26]. Moreover, researchers have studied the sulfate deterioration resistance of concrete modified using fly ash (FA), ground granulated blast-furnace slag (GGBS), silica fume (SF), and nanoparticles subjected to a sulfate environment [27,28,29,30,31]. Incorporation of the mineral additions into concrete has become one of the effective methods to eliminate the negative effects of sulfate deterioration [32]. Additionally, combined actions of SA and DW cycles have been paid increasing attention by a lot of researchers. Jiang and Niu [33] investigated the deterioration process of concrete that suffered DW cycles in different types of sulfate solutions. Aye and Oguchi [34] studied the sulfate resistance of plain and blended cement mortar containing pozzolans in sulfate solutions under full or partial immersion and DW cycles. Tian and Han [35] analyzed the relationship between the damage mechanism and microstructure of concrete under SA and DW cycles. Wang et al. [36] investigated the deterioration resistance coefficient (DRC) of compressive strength of concrete with mineral additions exposed to SA and DW cycles. It has been reported that the drying process removes the water from pores and hydrates, and leads to microcracks and shrinkage [37]. Though the rewetting regains water, the changes that occurred during the previous drying process are irreversible [38]. According to the literature review, most of the previous studies focused on the damage mechanism at the micro-level or the changes of mechanical parameters, the degradation law under coupled actions of SA and DW cycles remains unclear. The study of the influence of tunnel lining concrete mix proportions on degradation law is lacking. Particularly, the effect of influential factors such as sulfate concentration, water/binder (w/b) ratio, mineral addition, and curing regime on the degradation law of tunnel lining concrete under combined SA and DW cycles was limited in the literature.

Further, for a given concrete structure, its anticipated service life is dependent on material properties, environmental conditions, and maintenance practices [39]. The service life prediction of concrete under aggressive environments is receiving increasing attention. Various service life predictive models have been developed around the world, such as DuraCrete model [40], Life-365 model [41], EN 206-1 [42], the Scandinavian model “Clinconc” [43], a modified Fib2012-M model [44], and the “Exp-Ref” model [45]. Most service life prediction models are based on calcium leaching, chloride diffusion, and reinforcement corrosion [46,47,48,49,50,51,52,53,54,55,56,57]. Few studies have been made on service life prediction from the aspect of concrete strength damage. As we know, durability problems of the concrete structures usually appear as the material’s degradation at the beginning, then progressively lead to a reduction of the structural capacity, finally resulting in structural failure. Hence, studying the durability and service life prediction based on the degradation law of concrete strength is of great importance.

The main objective of this paper is to investigate the degradation law and service life prediction of the tunnel lining concrete with different mix proportions under SA and DW cycles. First, the effects of mix proportions on DRC under SA and DW cycles are studied, and the degradation law is analyzed. Then, the effects of influential factors such as sulfate concentration, mineral admixture content, w/b ratio, and curing regime on DRC are investigated. Influence coefficients of different influential factors are analyzed. After that, a new service life prediction model based on damage mechanics is developed and validated, and the proposed model is employed to predict the lifetime of concrete with different mix proportions.

## 2. Experimental Program and Evaluation Method

### 2.1. Materials

In this study, Ordinary Portland cement (OPC) conforming to GB 175-2007 [58] having the physical properties as given in Table 1 was used as the cement. Class-Ⅱ FA, S95 GGBS, and SF conforming to the Chinese standard were used in this study as partial replacement, with the performance index as listed in Table 2. The chemical compositions of OPC and mineral admixtures are listed in Table 3. River sand was chosen as the fine aggregates and crushed limestone as coarse aggregates. The physical characteristics of aggregates are listed in Table 4. A polycarboxylate-type water reducer keeping concrete slump no less than 80 mm was employed.

### 2.2. Mix Proportions and Test Design

Seven tunnel lining concrete mixtures with a constant w/b ratio of 0.41 were designed for studying the DRC and degradation law under the coupled actions of SA and DW cycles. As per GB/T 50082-2009 [59], a 5% Na_2_SO_4_ was used as the SA source. The mix proportions and relevant compressive strength of the concrete are given in Table 5. The control mixture included only OPC, while the remaining mixtures were a blend in which a proportion of OPC was partially replaced with the mineral admixtures. For example, F20G30S5 represents 20%, 30%, and 5% replacement of OPC with FA, GGBS, and SF, respectively. After demolding, the specimens were cured under standard conditions (20 ± 1 °C and RH ≥ 95%) for 28 days according to GB/T 50082-2009 [59].

Influential factors such as sulfate concentration, mineral admixture content, w/b ratio, and curing regime were considered. The test design of different influential factors is shown in Table 6. The mix proportions of concrete with different w/b ratios are shown in Table 7. All of the specimens were prepared with the dimensions of 100 × 100 × 100 mm complying with Chinese standards [59,60].

### 2.3. Test Procedure

The immersion process and DW cycles comply with GB/T 50082-2009 [59]. A standard DW cycle (24 h) proceeded as follows: when the curing age of concrete specimens reached 28 d, they were immersed in 5% Na_2_SO_4_ solution for 15 h with a temperature of 25–30 °C (Figure 2a); Then, the specimens were dried naturally by 1 h in air. After that, they were dried for 6 h at a temperature of 80 ± 5 °C (Figure 2b). Finally, the specimens were cooled for 2 h in air at (25–30) °C. According to the Chinese standard [59], these DW cycles were completed when the compressive strength of the concrete decreases to 75% of the initial value or the number of DW cycles researched 150 cycles. In this paper, the instrument for measuring concrete compressive strength adopted the full-automatic universal testing machine produced by Zhejiang Luda Machinery Instrument Co., Ltd. (Shaoxing, China). To avoid biased results, the results were obtained from the averaged compressive strength of three replicate specimens for each case.

### 2.4. Deterioration Resistance Evaluation

The ratio of the compressive strength of concrete suffered combined SA and DW cycles to the compressive strength of concrete without cyclic sulfate corrosion at the same curing age was defined to evaluate the DRC
(1)$$K_{DRC}=\frac{f_{n}}{f_{0}}$$
where $K_{DRC}$ is the DRC; $f_{n}$ is the compressive strength of concrete after *n* DW cycles; $f_{0}$ is the compressive strength of concrete cured under standard condition for 28 days; and *n* is the number of DW cycles.

The effects of influential factors such as sulfate concentration, mineral admixture content, w/b ratio, and curing regime on DRC are defined as:(2)$$I_{SC}=\frac{f_{SC}}{f_{SC\_5}}$$
(3)$$I_{MC}=\frac{f_{MC}}{f_{MC\_OPC}}$$
(4)$$I_{WB}=\frac{f_{WB}}{f_{WB\_0.41}}$$
(5)$$I_{CR}=\frac{f_{CR}}{f_{CR\_SC}}$$
where $I_{SC}$, $I_{MC}$, $I_{WB}$, and $I_{CR}$ are the influence coefficients of sulfate concentration, mineral admixture content, w/b ratio, and curing regime, respectively; $f_{SC}$, $f_{MC}$, $f_{WB}$, and $f_{CR}$ are the compressive strength of concrete considering influential factors; $f_{SC\_5}$ is the compressive strength of concrete subjected to 5% Na_2_SO_4_ solution; $f_{MC\_OPC}$ is the compressive strength of the 100% OPC concrete; $f_{WB\_0.41}$ is the compressive strength of the concrete with a w/b ratio of 0.41; and $f_{CR\_SC}$ is the compressive strength of concrete under standard curing. It is worth noting that all the specimens employed for assessing the influence coefficient were cycled 120 times.

## 3. Results and Discussion

### 3.1. Deterioration Resistance Coefficient of Concrete with Different Mix Proportions

Figure 3 presents the evolution of $K_{DRC}$ of concrete with different mix proportions subjected to DW cycles in 5% Na_2_SO_4_ solution. The corresponding fitting functions are given in Table 8. It can be seen that the $K_{DRC}$ first increases and then decreases. The increase is mainly owed to refinement effects due to the formation of expansive products in the core of the examples, while the degradation is due to cracking caused by the continuous formation of expansive compounds [61,62]. Similar deterioration behavior of compressive strength has already been reported by several researchers [36,63,64,65,66]. Additionally, it is evident that the concrete without any mineral admixtures (100% OPC concrete) has the most aggressive influence on the $K_{DRC}$ (i.e., down to 75% of original value after 90 DW cycles). The $K_{DRC}$ of concrete containing single mineral admixture drops faster than the composite mineral admixture after reach the peak value, and the concrete of F20G30S5 has the lowest decline rate of the $K_{DRC}$. The results suggest that the composite mineral admixture (F10G40, F20G30, and F20G30S5) had a more effective influence on $K_{DRC}$ compared to single mineral admixture (F20, G30, and S5).

### 3.2. Effect of Different Influential Factors on Deterioration Resistance Coefficient

#### 3.2.1. Effect of Sulfate Concentration

Figure 4 illustrates the effect of sulfate concentration on the DRC of the concrete containing different mineral admixtures subjected to DW cycles with SA. For the reference concentration, 5% Na_2_SO_4_ solution was chosen. As shown in Figure 4a–c, the influence coefficients of the OPC concrete ($I_{SC,OPC}$) and concrete with FA and GGBS ($I_{SC,FA}$ and $I_{SC,GGBS}$) are linearly decreased with the increase of sulfate concentration. When compared to the concrete immersed in tap water (i.e., concentration of 0%), the concrete with 100% OPC, FA, and GGBS showed a final influence coefficient loss of 29.2%, 26.2%, and 28.3% (15% solution), respectively. In Figure 4d, the influence coefficient of the concrete with SF ($I_{SC,SF}$) exhibits two distinct stages with the increase of the sulfate concentration, a first increase by 19.3% before 10% solution and a subsequent decrease to 1.0% at the end of DW cycles. The results indicate that the addition of SF displays a more effective improvement than FA and GGBS to enhance the sulfate resistance with the increase of sulfate concentration.

By using regression analysis, the effect of the sulfate concentration on DRC were established according to the following equations:

For the concrete without any mineral admixtures:

Concrete with 100% OPC:(6)$$I_{SC,OPC}=-0.0204SC+1.0995,\;R^{2}=0.986$$

For the concrete with a single mineral admixture:

Concrete with FA:(7)$$I_{SC,FA}=-0.0177SC+1.1235,\;R^{2}=0.916$$

Concrete with GGBS:(8)$$I_{SC,GGBS}=-0.0208SC+1.1273,\;R^{2}=0.959$$

Concrete with SF:(9)$$I_{SC,SF}=-0.0029SC^{2}+0.0438SC+0.8924,\;R^{2}=0.889$$

For the concrete with composite mineral admixture, a weighted average equation was defined as follows:(10)$$I_{SC,COMP}=\omega_{FA}I_{SC,FA}+\omega_{GGBS}I_{SC,GGBS}+\omega_{SF}I_{SC,SF}$$
where $I_{SC,OPC}$, $I_{SC,FA}$, $I_{SC,GGBS}$, and $I_{SC,SF}$ are the influence coefficients of the concrete with 100% OPC, FA, GGBS, and SF exposed to different sulfate concentrations, respectively; $I_{SC,COMP}$ is the influence coefficient of concrete with composite mineral admixture; *SC* in % is the sulfate concentration; $\omega_{FA}$, $\omega_{GGBS}$, and $\omega_{SF}$ are weight values determined as the ratio of FA, GGBS, and SF content to the total mineral admixture content (FA+GGBS+SF).

#### 3.2.2. Effect of Mineral Admixture Content

Figure 5 presents the effect of mineral admixture content on DRC of concrete exposed to DW cycles with SA. For the concrete with different FA contents, the influence coefficient ($I_{MC,\;FA}$) first increases and then decreases with the increase of the content, as shown in Figure 5a. The $I_{MC,\;FA}$ reaches the peak value (1.318) when the FA content is about 20%, and then gradually decreases to 1.227 when the FA content reaches 35%. A similar trend is also observed in GGBS concrete (Figure 5b). The influence coefficient ($I_{MC,GGBS}$) increases to 1.147 when GGBS content reaches 30%, then decreases to 1.088 when GGBS content reaches 40%. In Figure 5c, for the concrete with added SF, by increasing the SF content, the $I_{MC,SF}$ first decreases slightly, then increases slowly, and finally decreases gradually. The results indicate that the addition of FA in concrete displays a more effective improvement than GGBS and SF to enhance the sulfate resistance.

The effect of mineral admixture content on DRC were established as follows:(11)$$\mathrm{FA}:\;I_{MC,\;\;FA}=-0.0006C_{FA}^{2}+0.027C_{FA}+1,\;R^{2}=0.951$$
(12)$$\mathrm{GGBS}:\;I_{MC,GGBS}=-0.0002C_{GGBS}^{2}+0.0112C_{GGBS}+1,\;R^{2}=0.942$$
(13)$$\mathrm{SF}:\;I_{MC,SF}=-0.001C_{SF}^{3}+0.0172C_{SF}^{2}-0.0572C_{SF}+1,\;R^{2}=0.823$$
where $I_{MC,\;FA}$, $I_{MC,GGBS}$, and $I_{MC,SF}$ are the influence coefficients of concrete with different FA, GGBS, and SF contents; $C_{FA}$, $C_{GGBS}$, and $C_{SF}$ in % are the contents of FA, GGBS, and SF.

#### 3.2.3. Effect of Water–Binder Ratio

Figure 6 shows the effect of the w/b ratio on DRC of concrete under DW cycles with sulfate solution. For each w/b ratio, five sulfate concentrations (0%, 2.5%, 5%, 10%, and 15%) were considered, and the results were averaged. It is apparent that the influence coefficient ($I_{WB}$) linearly decreases with the increase of w/b ratio. A higher w/b ratio can increase the initial porosity of concrete, which leads to the easier entry of sulfate ions into the concrete, thus accelerating the deterioration process and causing more loss of concrete strength [28,65,67]. Additionally, for a given w/b ratio, the results also indicate that the $I_{WB}$ is insensitive to the sulfate concentration.

The effect of the w/b ratio on DRC were fitted as follows:(14)$$I_{WB}=-2.302\left(\frac{w}{b}\right)+1.971,\;R^{2}=0.963$$

#### 3.2.4. Effect of Curing Regime

Figure 7 shows the effect of the curing regime on the DRC of the concrete exposed to DW cycles with SA. The concrete with different FA and GGBS contents was studied, and their results were averaged. Clearly, the concrete under standard curing has the maximum $I_{CR}$ and the $I_{CR}$ of different curing regimes were as follows: SC (1.0) > FC (0.952) > WC (0.869) > SCC (0.791). As we know, the difference of the $I_{CR}$ is mainly due to the different environmental conditions (temperature and humidity) caused by the curing conditions. The strength and durability of concrete depend on the hydration of the cement. Proper curing could provide the appropriate temperature and humidity to ensure the hydration reactions process that produces the hydrated compounds to reduce the porosity [68].

### 3.3. Service Life Prediction Model under Sulfate Attack and Drying–Wetting Cycles

#### 3.3.1. Establishment of Life Prediction Model Based on Damage Mechanics

Considering the concept of one-dimensional degradation law as proposed [69,70], the degradation function of the concrete strength can be obtained from:(15)$$S=S_{0}\left(1-D\right)$$
where *S* and *S*_0_ are the current and initial strength values of a given material; *D* is the damage parameter represents the damage degree ranging between 0 (undisturbed state) and 1 (fully disturbed state). Here, *D* can be expressed as
(16)$$D=\frac{f_{0}-f_{n}}{f_{0}}=1-K_{DRC}\left(n\right)$$

According to the test results in the present study (Section 3.1), the relationship between damage parameter and the number of DW cycles can be given by
(17)$$-\frac{d\left(1-D\right)}{dn}\propto \left(1-D\right)$$

For the concrete with a given mix proportion, the slope of deterioration function is assumed to be a function related to the number of DW cycles, Equation (17) can be rewritten as:(18)$$\frac{d\left(1-D\right)}{dn}=-\lambda \left(n\right)\left(1-D\right)$$

Integration of Equation (18) leads to
(19)$$\left(1-D\right)=Ce^{-\int \lambda \left(n\right)dn}$$

Experimental results indicate that the compressive strength degradation conforms to parabolic law under combined actions of SA and DW cycles. Similar trends could also be found in previous works [36,63,64,65,66]. Hence, the λ(n) can be expressed as
(20)λ(n)=A+Bn
where *A* and *B* are constants determined by the mix proportions and the corrosion environment.

By substituting the expressions given in Equation (20) into Equation (19)
(21)$$\left(1-D\right)=Ce^{-\left(An+Bn^{2}\right)}$$

The boundary condition is as follows: when n=0, $f_{n}=f_{0}$. Based on Equation (16), Equation (21) is rewritten as
(22)$$K_{DRC}\left(n\right)=\frac{f_{n}}{f_{0}}=e^{-\left(An+Bn^{2}\right)}$$

It is evident that the degradation law of Equation (21) is the same as the fitting functions listed in Table 8.

Let us assume that the concrete subjected to sulfate environment for one year is equivalent to the DW cycles in a laboratory sulfate environment for *m* times. Hence, the corrosion time of concrete in a sulfate environment can be defined as t=n/m. Substituting it into Equation (22), the expression of $K_{DRC}\left(t\right)$ is obtained
(23)$$K_{DRC}\left(t\right)=e^{-\left[A\frac{t}{m}+B{\left(\frac{t}{m}\right)}^{2}\right]}$$
where $K_{DRC}\left(t\right)$ is the DRC at time *t*; *m* is the parameter related to the deterioration rate of the concrete in the real sulfate environment.

According to the practice of Glasser et al. [71], the service life prediction model of the tunnel lining concrete containing different influential factors can be expressed as follows
(24)$$K_{f}\left(t\right)=I_{SC}I_{MC}I_{WB}I_{CR}K_{DRC}\left(t\right)$$
where $K_{f}\left(t\right)$ is the DRC of concrete considering different influential factors subjected to SA and DW cycles at time *t*; $I_{SC}$ is the influence coefficient of sulfate concentration, which can be calculated by Equations (6)–(10); $I_{MC}$ is the influence coefficient of mineral admixture content, which can be calculated by Equations (11)–(13); $I_{WB}$ is the influence coefficient of w/b ratio, which can be calculated by Equation (14); $I_{CR}$ is the influence coefficient of curing regime, which can be taken according to the results in Section 3.2.4.

In this paper, we define the lifetime of concrete in the case of corrosion as the time when the initial strength decreases to 50% [72], namely the $K_{f}\left(t\right)$ reaches 0.5. In addition, it is noteworthy that the parameter *m* could be obtained by the mechanical test of the real existing concrete under sulfate environment. The flowchart of the service life prediction is summarized in Figure 8.

#### 3.3.2. Validation and Application of the Proposed Model

An application of the proposed prediction model was conducted on the concrete listed in Table 5. In this example, the parameter *m* was obtained by assuming the compressive strength of OPC concrete reduced by 50% after 100 years’ exposure. The w/b ratios of all the concrete were assumed to be 0.41, and the sulfate concentration was assumed to be 5%. The results are given in Table 9. The results indicate that the concrete without any mineral admixtures has the shortest lifetime (100 years for test and 102 years for prediction). Clearly, the concrete with composite mineral admixture has a longer lifetime than the concrete with a single mineral admixture. It is suggested that the composite mineral admixture has a more positive influence on the enhancement of the concrete lifetime than a single mineral admixture. Reference [73] also gives similar results. The mean absolute error between tests and predictions is about 9.9%. It implies that the predictive results agree well with the tests, though there are a few predictions with a higher error of nearly 15%. The difference might be attributed to experimental error, scattered property of concrete material, exposure conditions, and error of the model itself. Thus, it is reliable to employ the proposed model to predict the lifetime of concrete with different mix proportions. In this study, the degradation laws and influential factors were obtained by accelerated laboratory tests. It is necessary to conduct field tests to further validate and improve the model in the future.

## 4. Conclusions

Considering the coupled effects of SA and DW cycles, the DRC of the tunnel lining concrete with different mineral admixtures was investigated by experiments. The influence of sulfate concentration, mineral admixture content, w/b ratio, and curing regime on DRC was studied. The degradation law of the concrete was analyzed, and a new service life prediction model based on one-dimensional degradation law was developed. The main conclusions are summarized as follows:

(1) The concrete without any mineral admixtures (100% OPC concrete) has the most aggressive influence on the
$K_{DRC}$ under SA and DW cycles. The
$K_{DRC}$ of all concrete examples first increases and then decreases with the increase of DW cycles. The composite mineral admixture has a more positive influence on
$K_{DRC}$ compared to a single mineral admixture.

(2) The influence coefficients of the OPC, FA, and GGBS concrete are linearly decreased with the increase of sulfate concentration, while the concrete with addition SF exhibits a first increase and a later decrease. Additionally, the results indicate that the addition of SF in concrete displays a more effective improvement than FA and GGBS to enhance the sulfate resistance with the increase of sulfate concentration.

(3) For the concrete with different FA and GGBS contents, the influence coefficients of
$I_{MC,\;FA}$ and
$I_{MC,GGBS}$ first increase and then decrease with the increase of the content. For the concrete with SF, by increasing the SF content, the
$I_{MC,SF}$ first decreases slightly, then increases slowly, and finally decreases gradually. The results indicate that, with the increase of mineral admixtures content, the concrete with FA performs better sulfate resistance than the concrete with GGBS and SF.

(4) The influence coefficient of the w/b ratio (
$I_{WB}$) linearly decreases with the increase of the w/b ratio. For a given w/b ratio, the
$I_{WB}$ is insensitive to the sulfate concentration.

(5) The concrete under standard curing has the maximum
$I_{CR}$, and the
$I_{CR}$ of different curing regimes follows: SC (1.0) > FC (0.952) > WC (0.869) > SCC (0.791).

(6) The proposed service life prediction model was verified to be reliable. Additionally, it is suggested that the mineral admixtures can effectively improve the lifetime of the concrete. The composite mineral admixture has a more effective influence on the enhancement of concrete lifetime than a single mineral admixture.

## Figures and Tables

**Figure 1 materials-15-04435-f001:**
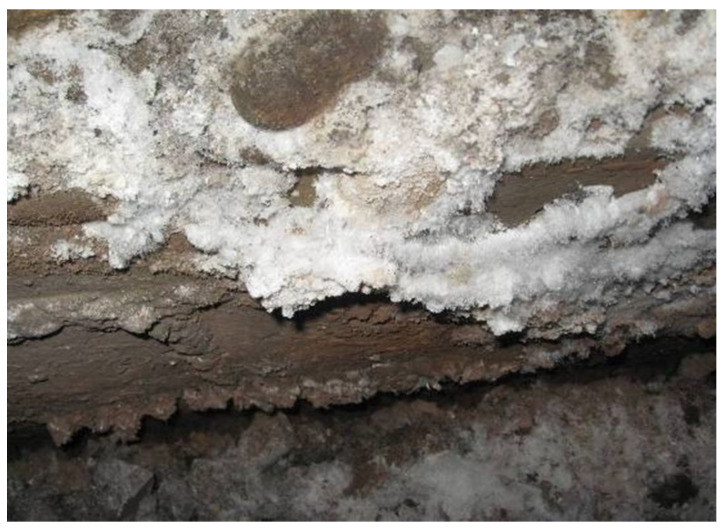
Sulfate attack of a tunnel lining in southwest China.

**Figure 2 materials-15-04435-f002:**
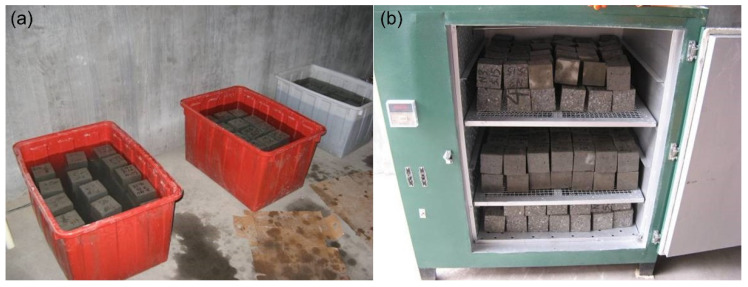
DW process with sulfate attack: (**a**) immersing; (**b**) drying.

**Figure 3 materials-15-04435-f003:**
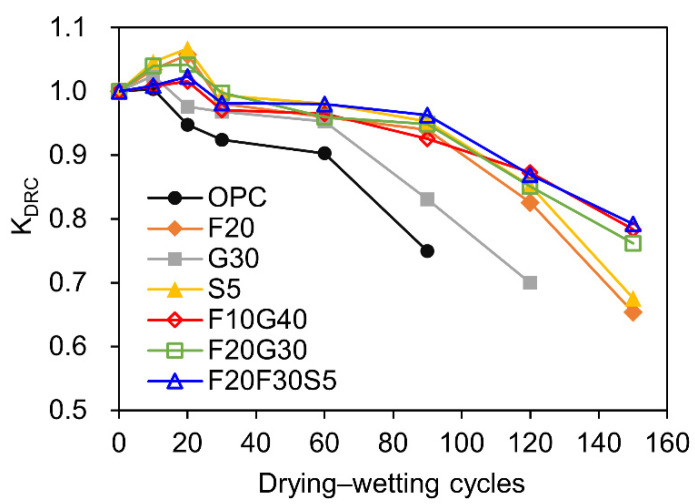
Evolution of $K_{DRC}$ of concrete with different mix proportions under DW cycles with SA (5% Na_2_SO_4_ solution).

**Figure 4 materials-15-04435-f004:**
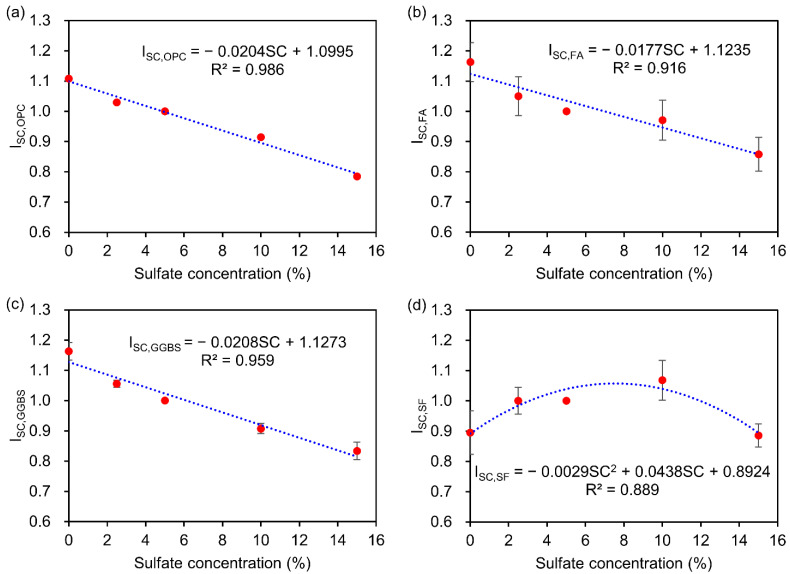
Effect of sulfate concentration on DRC of concrete containing different mineral admixtures exposed to DW cycles with SA: (**a**) OPC concrete; (**b**) FA concrete; (**c**) GGBS concrete; (**d**) SF concrete.

**Figure 5 materials-15-04435-f005:**
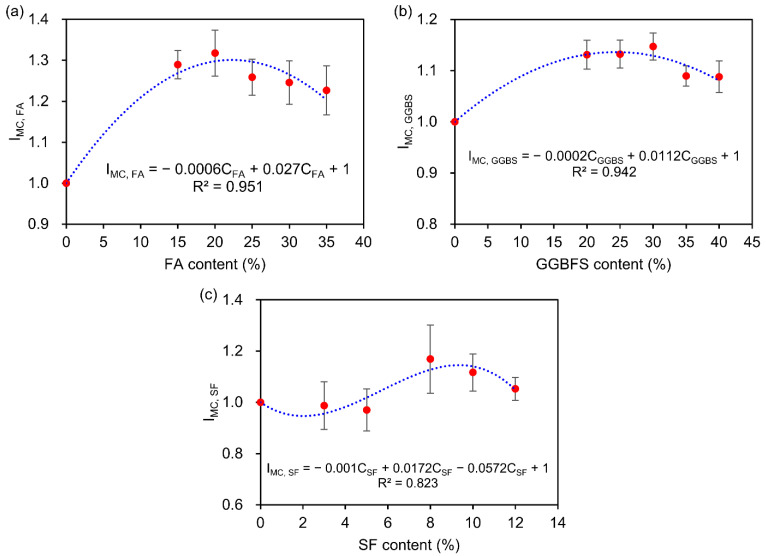
Effect of the mineral admixture content on DRC: (**a**) FA content; (**b**) GGBS content; (**c**) SF content.

**Figure 6 materials-15-04435-f006:**
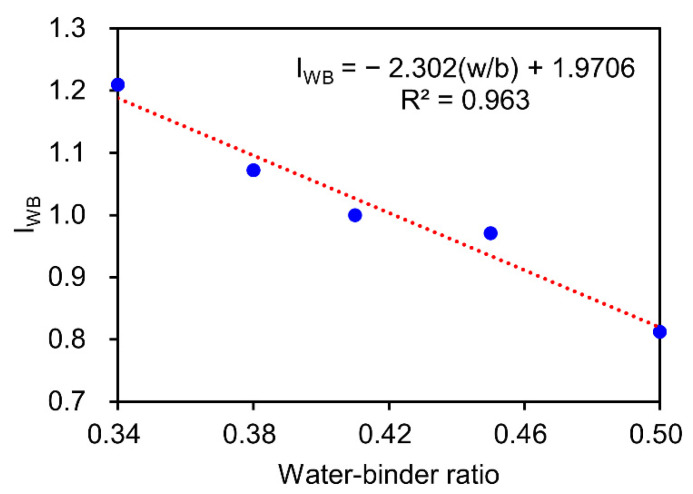
Effect of the w/b ratio on DRC of the concrete exposed to sulfate solution under DW cycles.

**Figure 7 materials-15-04435-f007:**
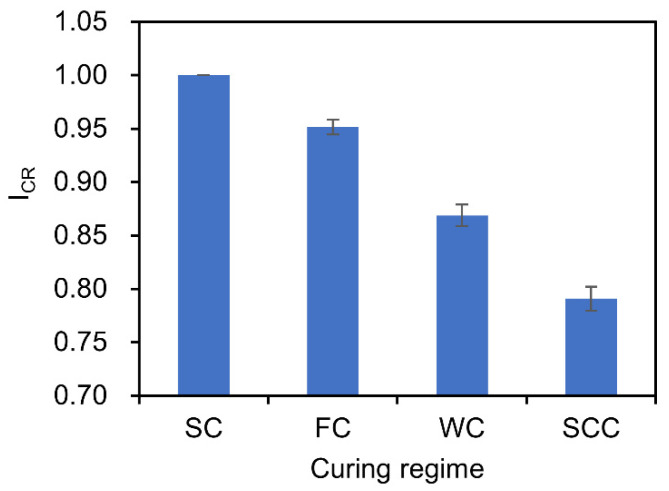
Effect of the curing regime on DRC.

**Figure 8 materials-15-04435-f008:**
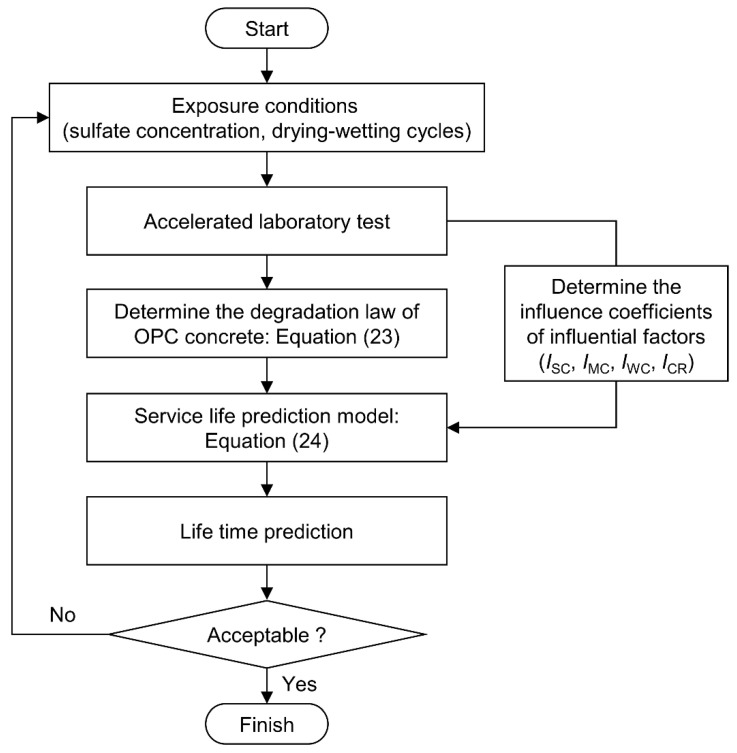
Flowchart for service life prediction of concrete subjected to DW cycles with SA.

**Table 1 materials-15-04435-t001:** Physical and mechanical properties of OPC used for this research.

Water Requirement of Normal Consistency (%)		Soundness		Loss on Ignition (%)	45 μm Sieving Residue (%)		Specific Surface Area (m^2^/kg)
28.4		Qualified		3.13	0.8		380
Setting time (min)		Flexural strength (MPa)			Compressive strength (MPa)		
Initial set	Final set	1 d	3 d	28 d	1 d	3 d	28 d
155	215	2.2	5.0	9.5	7.9	30.8	45.6

**Table 2 materials-15-04435-t002:** Performance index of mineral admixtures.

Mineral Admixtures	Water Requirement Ratio (%)	Loss on Ignition (%)	Moisture Content (%)	45 μm Sieving Residue (%)	Activity Index (%)		Specific Surface Area (m^2^/kg)
					7 d	28 d	
FA	0.98	1.57	0.6	1.2	65	87	650
GGBS	0.95	0.61	0.5	2.0	79	111	450
SF	-	1.66	0.9	1.0	-	103	18,000

**Table 3 materials-15-04435-t003:** Chemical compositions of cementitious materials (wt.%).

Materials	SiO_2_	CaO	Al_2_O_3_	Fe_2_O_3_	MgO	SO_3_	Na_2_O	K_2_O	f-CaO
Cement	21.09	62.50	4.34	2.81	1.81	2.87	0.15	0.62	0.67
FA	58.58	1.73	22.97	4.69	4.72	0.42	1.54	2.52	0.48
GGBS	36.61	39.47	11.79	1.23	8.94	0.23	-	-	-
SF	91.81	0.09	1.05	1.17	1.24	0.30	0.22	0.93	-

**Table 4 materials-15-04435-t004:** Physical characteristics of aggregates.

	Apparent Density (kg/m^3^)	Bulk Density (kg/m^3^)	Crushing Value Index (%)	Maximum Size (mm)	Fineness Modulus	Porosity (%)	Dust Content (%)
Fine aggregates	2590	1420	-	4.75	2.8	40.5	0.75
Coarse aggregates	2760	1450	8.8	25	-	42	0.1

**Table 5 materials-15-04435-t005:** Mixture proportions and compressive strength of concrete.

Mix ^a^	Cement (kg/m^3^)	Mineral Admixture Content (%)			Coarse Aggregates (kg/m^3^)	Fine Aggregates (kg/m^3^)	Water (kg/m^3^)	Compressive Strength (MPa)			
		FA	GGBS	SF				3 d	7 d	28 d	56 d
OPC	395	-	-	-	1032	811	162	30.8	45.6	50.5	53.0
F20	316	79	-	-	1032	811	162	28.2	38.6	53.8	57.2
G30	276.5	-	118.5	-	1032	811	162	26.2	43.4	54.9	58.2
S5	375.25	-	-	19.75	1032	811	162	28.8	38.2	52.7	57.4
F10G40	197.5	39.5	158	-	1032	811	162	29.2	44.7	54.2	60.3
F20G30	197.5	79	118.5	-	1032	811	162	27.0	40.7	59.1	63.4
F20G30S5	177.75	79	118.5	19.75	1032	811	162	30.2	44.9	58.4	62.6

^a^: OPC represents the content of cement is 100%; F represents FA; G represents GGBS; S represents SF. The number following the letter is the percentage of mineral admixtures used to replace the OPC. All of the concretes were prepared with a w/b ratio of 0.41.

**Table 6 materials-15-04435-t006:** Test design of different influential factors.

Influential Factors	Description	Values or Conditions
sulfate concentration (%)	-	0, 2.5, 5, 10, 15
mineral addition content (%)	FA	0, 15, 20, 25, 30, 35
	GGBS	0, 20, 25, 30, 35, 40
	SF	0, 3, 5, 8, 10, 12
w/b ratio ^a^	-	0.34, 0.38, 0.41, 0.45, 0.5
curing regime	standard curing (SC)	20 ± 1 °C and RH ≥ 95%
	fog curing (FC)	20 ± 5 °C and RH ≥ 90%
	water curing (WC)	22 ± 5 °C
	same condition curing (SCC)	22 ± 5 °C and RH ≥ 65%

^a^: The control w/b ratio was 0.41. The mix proportions of concrete with different w/b ratios are shown in Table 7. Moreover, five sulfate concentrations (0%, 2.5%, 5%, 10%, and 15%) were considered for each w/b ratio.

**Table 7 materials-15-04435-t007:** Mix proportions of concrete with different w/b ratios.

w/b (%)	Cement (kg/m^3^)	Fine Aggregates (kg/m^3^)	Coarse Aggregates (kg/m^3^)	Superplasticizer (%)
0.34	470	710	1110	1.15
0.38	420	738	1107	1.0
0.41	395	774	1069	0.95
0.45	350	803	1064	1.0
0.50	310	829	1056	1.1

**Table 8 materials-15-04435-t008:** Fitting parameter and R-square of the empirical formulas for DRC.

Mix Proportions	Fitting Functions	R-Square
OPC	$K_{DRC}=e^{-1.1572\times 10^{-3}n-2.0402\times 10^{-5}n^{2}}$	0.935
F20	$K_{DRC}=e^{1.5437\times 10^{-3}n-2.8055\times 10^{-5}n^{2}}$	0.952
G30	$K_{DRC}=e^{5.5630\times 10^{-4}n-2.9112\times 10^{-5}n^{2}}$	0.975
S5	$K_{DRC}=e^{2.0192\times 10^{-3}n-3.0053\times 10^{-5}n^{2}}$	0.948
F10G40	$K_{DRC}=e^{8.8134\times 10^{-4}n-1.8192\times 10^{-5}n^{2}}$	0.956
F20G30	$K_{DRC}=e^{1.7376\times 10^{-4}n-1.1776\times 10^{-5}n^{2}}$	0.976
F20G30S5	$K_{DRC}=e^{6.8270\times 10^{-4}n-1.4922\times 10^{-5}n^{2}}$	0.969

**Table 9 materials-15-04435-t009:** Service life prediction results.

Mix Proportion	Regression Constants of $K_{DRC}\left(t\right)=e^{-\left[A\frac{t}{m}+B{\left(\frac{t}{m}\right)}^{2}\right]}$		M ^a^	$K_{f}\left(100\right)$	Lifetime (Year)		
	A	B			Tested ^b^	Predicted ^c^	Error ^d^ (%)
OPC	1.1572 × 10^−3^	2.0402 × 10^−5^	0.632	0.5	100	102	2.0
F20	−1.1537 × 10^−3^	2.8055 × 10^−5^	0.632	0.5	113	124	9.8
G30	−5.5630 × 10^−4^	2.9112 × 10^−5^	0.632	0.5	104	115	11.0
S5	−2.0192 × 10^−3^	3.0053 × 10^−5^	0.632	0.5	120	107	−10.7
F10G40	−8.8134 × 10^−4^	1.8192 × 10^−5^	0.632	0.5	140	127	−9.0
F20G30	−1.7376 × 10^−4^	1.1776 × 10^−5^	0.632	0.5	158	134	−15.5
F20G30S5	−6.8270 × 10^−4^	1.4922 × 10^−5^	0.632	0.5	152	135	−11.0

^a^: The parameter *m* was calculated assuming the compressive strength of the OPC concrete shown in Table 5 reduced by 50% after 100 years’ cyclic sulfate corrosion in 5% Na_2_SO_4_ solution. ^b^: The service lifetime was predicted based on the corresponding regression constants of the seven concrete mixtures listed in the first column in the table. ^c^: The service lifetime was predicted based on the regression constants of OPC concrete (i.e., A = 1.1572 × 10^−3^ and B = 2.0402 × 10^−5^). ^d^: Error (%) = ((Predicted lifetime − Tested lifetime)/Tested lifetime) × 100.

## Data Availability

Not applicable.

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
