# Peer review of "Degradation Law and Service Life Prediction Model of Tunnel Lining Concrete Suffered Combined Effects of Sulfate Attack and Drying–Wetting Cycles"

_materials, 2022, doi:10.3390/ma15134435_

Round 1

Reviewer 1 Report

The manuscript „Degradation law and service life prediction model of tunnel lining concrete suffered combined effects of sulfate attack and dry-wetting cycles“ compares the efficiency of various mineral admixtures and some influential factors towards sulfate attack and drying-wetting cycles. There are interesting information specially about service life prediction, but overall the research has brought more questions than it has answered. I recommend to done some analysis for answering some questions and confirming some statements.

Before publishing I would recommend the following points to the attention of the authors:

  • Why are in the table 5 chosen exactly these 7 mixtures. Why you did not published the compressive strength of other mixtures?
  • Please add the specification of the instrument for measuring compressive strength.
  • Line 175 – Specify the expansive products and support it with relevant analyses.
  • 4 d – How you explain the increase and decrease of Isc,SF , that is totally different than other mineral admixtures.
  • 6 – add the error bars
  • Line 344 – 348 – Explain why the admixtures has positive influence on KDCR
  • Conclusions 2 and 3 – You contradict in these conclusions. First you said, that SF is more effective to enhance the sulfate resistance and than you said FA more effective to enhance the sulfate resistance.

Reviewer 2 Report

The research topic is interesting and findings could be useful in the industry. The only suggestion to improve the paper is to increase citations in discussion. Justifications must be supported by references and more comparison studies should be added.

Most of the references (almost 85%) were used in the introduction and research methodology. In a good research paper, it is expected at least 30% of listed references to be used in results and discussion. The authors can use references to justify the results and also for comparison purposes. For example, the statement in line 317-318, the claim of “concrete with composite mineral admixture has a longer lifetime” should be supported by previously published books and/or papers.

Reviewer 3 Report

1.More and newer findings from other researchers to be included in the introduction.

2.The findings should be compared with those of others in the discussion section.

3.The manuscript concludes that SF concrete displays a more effective improvement than FA and GGBS concretes. But Figure 3 shows the same behavior for SF concrete and FA concrete.

4.Most references are old and should be replaced with newer ones.

Round 2

Reviewer 1 Report

In the manuscript are some interesting information specially about service life prediction, but lot of predictions are not supported by relevant analysis. In the end I recommend to publish this manuscript.